GDF11 induces differentiation and apoptosis and inhibits migration of C17.2 neural stem cells via modulating MAPK signaling pathway

Wang Zongkui
Dou Miaomiao
Liu Fengjuan
Jiang Peng
Ye Shengliang
Ma Li
Cao Haijun
Du Xi
Sun Pan
Su Na
Lin Fangzhao
Zhang Rong kylie2009@foxmail.com
Li Changqing lichangqing268@163.com
Institute of Blood Transfusion, Chinese Academy of Medical Sciences & Peking Union Medical College , Chengdu , Sichuan , China
Mourtada-Maarabouni Mirna
Electronic publication date: 2018 Sep 4
Publication date: 2018
Volume: 6
Electronic Location ID: e5524
Received 2018 Jun 19; Accepted 2018 Aug 7
Copyright: ©2018 Wang et al.
Copyright year: 2018
Copyright holder: Wang et al.
License: This is an open access article distributed under the terms of the Creative Commons Attribution License, which permits unrestricted use, distribution, reproduction and adaptation in any medium and for any purpose provided that it is properly attributed. For attribution, the original author(s), title, publication source (PeerJ) and either DOI or URL of the article must be cited.
License URL: https://creativecommons.org/licenses/by/4.0/

Keywords: Growth differentiation factor 11, C17.2 neural stem cells, Differentiation, Apoptosis, MAPK signaling pathway, Migration

Funding: PUMC Youth Fund and the Fundamental Research Funds 2017310036) CAMS Innovation Fund for Medical Sciences 2016-I2M-1-018 2017-I2M-3-021 Scientific Research Project of Sichuan Medical Association S17082 This work was supported by the PUMC Youth Fund and the Fundamental Research Funds for the Central Universities (grant number 2017310036), the CAMS Innovation Fund for Medical Sciences (grant number 2016-I2M-1-018 and 2017-I2M-3-021), and the Scientific Research Project of Sichuan Medical Association (grant number S17082). The funders had no role in study design, data collection and analysis, decision to publish, or preparation of the manuscript.

==============================
GDF11, a member of TGF-β superfamily, has recently received widespread attention as a novel anti-ageing/rejuvenation factor to reverse age-related dysfunctions in heart and skeletal muscle, and to induce angiogenesis and neurogenesis. However, these positive effects of GDF11 were challenged by several other studies. Furthermore, the mechanism is still not well understood. In the present study, we evaluated the effects of GDF11 on C17.2 neural stem cells. GDF11 induced differentiation and apoptosis, and suppressed migration of C17.2 neural stem cells. In addition, GDF11 slightly increased cell viability after 24 h treatment, showed no effects on proliferation for about 10 days of cultivation, and slightly decreased cumulative population doubling for long-term treatment (p < 0.05). Phospho-proteome profiling array displayed that GDF11 significantly increased the phosphorylation of 13 serine/threonine kinases (p < 0.01), including p-p38, p-ERK and p-Akt, in C17.2 cells, which implied the activation of MAPK pathway. Western blot validated that the results of phospho-proteome profiling array were reliable. Based on functional analysis, we demonstrated that the differentially expressed proteins were mainly involved in signal transduction which was implicated in cellular behavior. Collectively, our findings suggest that, for neurogenesis, GDF11 might not be the desired rejuvenation factor, but a potential target for pharmacological blockade.

Introduction

Growth differentiation factor 11 (GDF11), also known as bone morphogenetic protein 11 (BMP11), is a secreted glycoprotein belonging to the transforming growth factor β (TGF-β) superfamily (Pepinsky et al., 2017; Walker et al., 2016). GDF11 plays an important role in anterior/posterior axial patterning during embryonic development (Oh et al., 2002). Similar to the negative effects of myostatin (also known as GDF8), which is 90% homology with GDF11 (Walker et al., 2017), in skeletal muscle, GDF11 acts as a negative regulator of neurogenesis in the olfactory epithelium (Wu et al., 2003) and in the developing spinal cord (Santos et al., 2012).

Recently, Loffredo et al. (2013) suggested that GDF11 was the rejuvenation factor to reverse age-related dysfunction in heart. Subsequently, it was confirmed that GDF11 repaired injured skeletal muscle and improved cognitive function (Katsimpardi et al., 2014; Sinha et al., 2014). However, the rejuvenating effects of GDF11 on heart, skeletal muscle and brain were questioned by a couple of independent studies. Egerman and colleagues (2015) showed that GDF11 supplementation inhibited muscle regeneration and decreased satellite cell expansion in mice, and suggested that GDF11 was not a rejuvenation factor but a potential target for pharmacologic blockade to treat age-related diseases. Hinken et al. (2016) also suggested GDF11 wasn’t a rejuvenator of aged murine skeletal muscle satellite cells. In addition, others reported restoring GDF11 in old mice had no effect on cardiac structure or function (Smith et al., 2015). These conflicting studies offer compelling evidence that the effects of GDF11 are contradictory and demonstrate that the effects of GDF11 on neurogenesis are still not completely understood. Therefore, we require an in-depth knowledge of the effects and potential mechanism of action of GDF11 on regulating neural stem cells.

In the present study, we focused on the effects of GDF11 on the cellular behavior of C17.2 neural stem cells (including viability, proliferation, differentiation, apoptosis and migration), the changes in the phospho-proteome and the corresponding signaling pathways. We herein showed that GDF11 promoted differentiation and apoptosis, and suppressed migration of C17.2 cells. In addition, GDF11 stimulated cellular proliferation in a short time (within 24 h), whereas high concentrations of GDF11 inhibited proliferation in a long-term cultivation (∼20 days). Pathway-oriented proteome profiling revealed that GDF11 stimulation significantly activated phosphorylation of 15 proteins, including Smad2/3, Erk1/2, Akt1/2/3, p38, p70S6k, GSK-3α/3β and HSP27, which were mainly involved in MAPK signaling pathway. These data demonstrated the effects of GDF11 on neural stem cell (inducing differentiation and apoptosis, and suppressing migration) through the MAPK pathway.

Materials and Methods

Agents

C17.2 neural stem cell line was purchased from zqxzbio (Shanghai, China). Dulbecco’s modified Eagle’s medium (DMEM; catalog No. SH30022.01), Penicillin-Streptomycin Solution (catalog No. SV30010) and Trypsin (catalog No. SH30236.01) were obtained from Hyclone (Logan, Utah, USA). Fetal bovine serum (FBS; catalog No. 04-121-1A-US) and horse serum (HS; catalog No. 04-124-1A) were purchased from Biological Industries (Beit Haemek, Israel). Recombinant human/mouse/rat GDF-11/BMP-11 (catalog No. 1958-GD-010) and Phospho-MAPK proteome profiler array kit (catalog No. ARY002B) were obtained from R&D systems (Minneapolis, MN, USA). Dimethyl sulfoxide (DMSO; catalog No. D2650) were purchased from Sigma-Aldrich (St. Louis, MO, USA). NuPAGE LDS loading buffer (catalog No. NP0007) and NuPAGE 4–12% Bis-Tris gel (catalog No. NP0321BOX) were obtained from Invitrogen (Waltham, MA, USA). Phosphatase inhibitor cocktail (catalog No.HY-K0022) and protease inhibitor cocktail (catalog No. HY-K0010) were from MedChem Express (Shanghai, China). RIPA buffer (catalog No. 89900) was from Pierce (Rockford, USA). Rabbit anti-Smad2/3 (catalog No. 8685S), rabbit anti-p-smad2/Smad3 (catalog No. 8828S) , rabbit anti-CREB (catalog No. 9197S), rabbit anti-p-CREB (catalog No. 9198S), rabbit anti-ERK (catalog No.12950S), rabbit anti-p-ERK (catalog No.4377T), rabbit anti-p38 (catalog No.8690S), rabbit anti-p-p38 (catalog No.4511T), mouse anti-nestin (catalog No.33475S), rabbit anti-βIII-tubulin (catalog No.5568S), rabbit anti-GFAP (catalog No.12389S), rabbit anti-β-actin (catalog No.4970S), rabbit anti-GAPDH (catalog No.2118S) and goat anti-rabbit IgG (catalog No.7074P2) were obtained from Cell Signal Technology (Beverly, MA, USA). Enhanced Cell Counting Kit-8 (CCK-8; catalog No.C0042) and BCA Protein Assay Kit (catalog No.P0010) were obtained from Beyotime Biotechnology (Beijing, China). Enhanced chemiluminescence (ECL; catalog No.32109) was form Pierce (Rockford, IL, USA).

C17.2 cell culture

C17.2 cells were cultured on 25-cm2 culture flasks in complete medium (DMEM supplemented with 10% (v/v) FBS, 5% (v/v) HS, 100 U/ml penicillin and 100 µg/ml streptomycin) at 37 °C, 5% CO2 in a humidified atmosphere. Media were changed every 2–3 days. After reaching 70–80% confluence, C17.2 cells were trypsinized and re-seeded at a density of 4*104 cells/mL in complete medium which was changed to starved medium (DMEM supplemented with 0.5% HS and 1% FBS) one day post seeding. After 6 h of serum starvation, different concentrations of GDF11 (0, 12.5, 25, 50 and 100 ng/mL) were added, respectively.

Cell morphology analysis

C17.2 cells were seeded onto the 24-well plates at a density of 4*104 cells/well in 0.1 mL complete medium. After adherence, complete medium was replaced with starved medium for 6 h and then, various concentrations of GDF11 were introduced when appropriate. GDF11-untreated cells were served as control.

The cell morphology and viability were examined using LIVE/DEAD® viability/cytotoxicity kit (catalog No.L3224) for mammalian cells (Invitrogen, Carlsbad, CA, USA) according to the manufacturer’s instructions under inverted fluorescence microscope (AXIO, Zeiss, Jena, Germany). The live cells were stained with calcein AM in green, and the dead cells were stained with ethidium homodimer-1(EthD-1) in red.

Cell viability and proliferation assay

Cell viability was assessed by CCK-8 assay. Briefly, 10 µL of CCK-8 agent was added to each well 2 h before the termination of the experiment. The optical density (OD) values at 450 nm were determined using SpectraMax M2e (Molecular Devices, Sunnyvale, USA). Then, by comparing the absorbance of GDF11-treated and untreated cells, percentage viability was calculated.

For proliferation assay, 1*104/mL cells were seeded in 12-well plates in triplicates. When the cell cultured to ∼80% confluence (generally 3 days), cells were trypsinized and manually counted using a haemocytometer. Cell population doubling (PD) was calculated using the following formulae: (1) PD= log2N∕N0,

where N0 represents the number of cells seeded at the initial passage, N is the final number of cells.

Apoptosis assay

To investigate the apoptosis-inducing effect of GDF11, we identified apoptotic and necrotic cells by Annexin V-FITC and propidium iodide (PI) dual staining using FACScan flow cytometry (Becton-Dickinson, Franklin Lakes, NJ, USA). Approximately 1*105 cells were analyzed in each experimental group. The cell populations were distinguished according to their positioning of quadrants: live cells (Annexin V−/PI−), early/primary apoptotic cells (Annexin V+/PI−), late/secondary apoptotic cells (Annexin V+/PI+) and necrotic cells (Annexin V−/PI+).

Scratch wound healing assay

C17.2 cells were cultured with complete medium in a 48-well plate at a density of 5 × 104 cells/well. After reaching ∼80% confluence, a single uniform scratch was made by using a 200 µL pipette tip along the center of each monolayer. The scratch was lightly washed with PBS twice to remove the detached cells, and then starved medium supplemented with various concentrations of GDF11 was added (0 ng/mL, 12.5 ng/mL, 25 ng/mL, 50 ng/mL and 100 ng/mL, respectively). The scratches were monitored at 0 h, 12 h and 36 h after scratching by taking photos with inverted microscope to measure the wound closure. The wound closures of various treatments at different time points were calculated with Image J software.

RNA extraction and qRT-PCR analysis

C17.2 cells were cultured on 12-well plates at a density of 4*104 cells per well under standard conditions. Upon reaching 80% confluence, the complete medium was changed to starved medium. After 6 h of serum starvation, plates were treated with either indicated concentrations of GDF11 (25 ng/mL, 50 ng/mL and 100 ng/mL, respectively) or vehicle in starved medium for 4 h. Total RNA was extracted from the cultured cells using TRIZOL reagent according to the standard procedure. Total RNA (1 µg) was reverse transcribed in a final volume of 20 µL in a reaction containing random primers, using iScriptTM cDNA Synthesis kit (Bio-Rad, Hercules, CA, USA). qRT-PCR was done using the Quantitect SYBR Green PCR kit (Qiagen, Valencia, CA, USA) with a ABI StepOnePlus Real-time PCR system (Applied Biosystems, Foster City, USA). Relative expression was calculated using the 2−ΔΔCt method by normalizing with GAPDH housekeeping gene expression and presented as fold changes relative to control. The primers for qRT-PCR were synthesized by Beijing Genomics Institute (Shenzhen, China) and the details of primer sequences are shown in Table S1.

Phospho-proteome profiling array

Human phospho-MAPK array kit was used to determine the relative levels of phosphorylation of mitogen-activated protein kinases (MAPKs) and other serine/threonine kinases with or without GDF11 treatment. Briefly, C17.2 cells were rinsed with PBS and solubilized with Lysis Buffer 6 (provided in Human Phospho-MAPK Array Kit) at 1*107 cells/mL. After rocking gently at 2–8 °C for 30 min, the lysates were centrifuged at 14,000 g for 5 min, and the supernatant was collected and detected the protein contents using BCA protein assay. The arrays were blocked by Buffer 5 for 1 h on a rocking platform shaker. Afterwards, the mixture of sample and detection antibody cocktail were introduced and incubated overnight at 2–8 °C on a rocking platform shaker. The following day, the membranes were washed three times, and then were incubated in streptavidin-HRP for 30 min followed by three washes. The protein blots were developed by ECL reagents. Densitometry analysis was measured with the Quantity One software, and the averaged intensity was calculated by subtracting the averaged background signal. The fold change was obtained by comparing GDF11-treated samples with the untreated control (indicated as a value of 1): Fold change=average intensityGDF11-treated∕average intensitycontrol.

The respective coordinates of all the antibodies on the arrays and the corresponding phosphorylation sites can be found in Table S2.

Western Blot analysis and validation

C17.2 cells were cultured in six-well dishes in starved medium with or without GDF11 for 24 h. Then, the cells were lysed in RIPA buffer containing 1 × phosphatase inhibitor cocktail and 1 × protease inhibitor cocktail on ice for 30 min, and centrifuged at 14,000 g for 5 min at 4 °C. The supernatants were collected, and the protein concentration was determined by BCA protein assay kit. The samples were mixed with 4 × NuPAGE LDS loading buffer, separated on NuPAGE 4–12% Bis-Tris gels, and subsequently transferred to PVDF membranes by a wet transfer apparatus (Bio-Rad, Hercules, CA, USA). Following blocking with superblock at room temperature for 2 h, the membranes were incubated with rabbit anti-β-actin (1:1,000), anti-Smad2/3 (1:1,000), anti-p-Smad2/3 (1:1,000), anti-CREB (1:1,000), anti-p-CREB (1:1,000), anti-ERK (1:1,000), anti-p-ERK (1:1,000), anti-p38 (1:1,000), anti-p-p38 (1:1,000), anti-nestin (1:1,000), anti-βIII-tubulin (1:1,000), anti-GFAP (1:1,000) and anti-GAPDH (1:1,000), respectively, at 4 °C overnight. After washing with PBST (PBS supplemented with 0.1% Tween 20), membranes were incubated with horseradish peroxidase-conjugated goat anti-rabbit IgG (1:3,000) at room temperature for 2 h, and were visualized by ECL reagents. ImageJ software (Schneider, Rasband & Eliceiri, 2012) was used for densitometric analyses of the blots.

Bioinformatic analyses

To further understand the functions and features of the identified and quantified proteins, we annotated functions and features of proteins from several different categories, including subcellular localization, domain, Gene Ontology (GO) and pathway.

WoLF PSORT (a subcellular localization predication tool, a version of PSORT/PSORT II) and iLoc-Animal (http://www.jci-bioinfo.cn/iLoc-Animal) were used to predict subcellular localization of all identified differentially expressed proteins.

The domain functional description of the differentially expressed proteins were annotated by InterProScan (a sequence analysis application) based on protein sequence alignment method, and the InterPro domain database was used (http://www.ebi.ac.uk/interpro/).

GO annotation was derived from the UniProt-GOA database (http://www.ebi.ac.uk/GOA/), and the differentially expressed proteins were classified by GO annotation based on the three categories (GO term level 1): biological process, cellular component and molecular function. According to GO annotation information of the identified proteins, we summed up the amount of the differentially expressed proteins in each GO term of level 2.

The protein–protein interaction networks and pathways were annotated by Kyoto Encyclopedia of Genes and Genomes (KEGG) database.

Statistical analysis

The results were presented as the mean ± standard error (SE). Multi-group comparisons were performed by one-way ANOVA followed by Tukey’s post hoc test. Paired analysis of control and treatment was accomplished using two-tailed unpaired or unpaired Student’s t –tests when appropriate. In addition, Statistical analyses were conducted using SPSS statistics software,version 17.0 (SPSS Inc., Chicago, IL, USA), and p < 0.05 was considered statistically significant.

Results

The positive and negative effects of GDF11 on cellular viability and proliferation

When compared with the counts of C17.2 cells initially seeded, both GDF11- and vehicle-treated cells significantly proliferated after 72 h of cultivation (Fig. S1). Imaging revealed that GDF11 significantly altered the morphology of C17.2 cells (Fig. 1A). Cells without GDF11 treatment remained their native neural stem cell state (Fig.1a, control), whereas cells treated with various concentrations of GDF11 showed visual outgrowth of neuritis, displaying phenotypes similar to neuron- and astrocyte-like cells (Fig. 1A, GDF11). Remarkably, compared to the control, supplement with high concentrations of GDF11 (50 and 100 ng/mL) significantly resulted in morphological changes (differentiation and apoptosis) (Fig. S1A).

Figure 1 Effect of GDF11 on C17.2 cells.

(A) The representative images of live and dead cell staining. C17.2 cells were cultured with indicated concentrations of GDF11. Images were obtained at 200× magnification by inverted fluorescence microscope. The live cells were stained with calcein AM in green, and the dead cells were stained with EthD-1 in red. (B) GDF11 induced apoptosis in C17.2 neural stem cells. C17.2 cells were treated with vehicle or GDF11 (12.5, 25, 50 and 100 ng/mL) for 48 h and cell distribution was analysed using Annexin V-FITC and PI dual staining. The FITC and PI fluorescence was measured by flow cytometer with FL-1 and FL-2 filters, respectively. Lower left quadrant–live cells (Annexin V−/PI−), lower right quadrant–early/primary apoptotic cells (Annexin V+/PI−), upper right quadrant–late/secondary apoptotic cells (Annexin V+/PI +) and upper left quadrant–necrotic cells (Annexin V−/PI+). (C) The viability of C17.2 cells after 24 h or 72 h of cultivation with various concentrations of GDF11 or vehicle was measured using CCK-8 method. N = 3, p < 0.05. (D) Cumulative population doubling levels of C17.2 cells supplemented with different GDF11 concentrations for a total period of 6 passages. N = 4, *p < 0.05 compared with control. (E) Quantitative analyses of the GDF11 effect on apoptosis. N = 3, *p < 0.05 versus with vehicle control and **p < 0.01 versus with vehicle control.

To investigate the effect of GDF11 on cell viability, C17.2 cells were treated with indicated concentrations of GDF11 (0, 12.5, 25, 50 and 100 ng/mL) for a 72 h period, followed by CCK-8 assays. GDF11 slightly increased (less than 10%, p < 0.05) cell viability after 24 h treatment, whereas it did not affect the cell viability after 72 h treatment (Fig. 1C).

As displayed in Fig. 1D, all groups of C17.2 cells showed robust proliferation for the six-passage duration. GDF11 showed no effect on C17.2 cell proliferation until the 4th passage. From the 5th passage, the low concentrations of GDF11 (12.5 and 25 ng/mL) still didn’t affect the proliferation of C17.2 cells, whereas higher concentrations of GDF11 (50 and 100 ng/mL) significantly inhibited cell proliferation (p < 0.05) and the exposure of C17.2 cells to 100 ng/mL GDF11 resulted in the lowest cumulative population doubling level during the 6 passages of cultivation amongst the five groups, which was approximately 17% lower than control (p < 0.05).

Next, we detected the mRNA expression of cyclin D1 and cyclin D2, the cell cycle-related proteins. GDF11 slightly but not significantly attenuated the expression of cyclin D1 and cyclin D2 in the mRNA levels (Fig. 2D; p > 0.05). These provide a potential molecular basis for the effects of GDF11 on C17.2 cell viability and proliferation.

Figure 2 The effect of GDF11 on mRNA and protein expression.

(A) Nestin, βIII-tubulin and GFAP mRNA levels of C17.2 cells after GDF11 or vehicle (control) treated for 5 h.The results display mean ±SD of n = 4 and were analysed by one-way ANOVA followed by Tukey’s post hoc test. *p < 0.05 as compared with mRNA levels in control cells. (B) Nestin, βIII-tubulin and GFAP protein levels of C17.2 cells after GDF11 (50 ng/mL, “T”) or vehicle (“C”) treated for 72 h. (C) Quantitative analyses of protein expression in relation to β-actin expression. Results were analysed by Student’s t-test. N = 6, *p < 0.05. (D) Cyclin D1, Cyclin D2 and EGFR mRNA expression after GDF11 or vehicle treated for 5 h. N = 5, *p < 0.05 and **p < 0.01. (E) The mRNA levels of Smad2, Smad3, Alk5 and ActRIIB after GDF11 or vehicle treated for 5 h. N = 5, *p < 0.05 and **p < 0.01.

Together, these results revealed that GDF11 slightly increased cell viability after a short-term (24 h) cultivation and showed no effect on cell viability from 1st to 4th passage of cultivation (approximately 10 days), whereas high concentrations of GDF11 significantly suppressed cumulative population doubling for a long-term treatment.

GDF11 induced differentiation and apoptosis of C17.2 cells

The mRNA levels of the neural progenitor cell marker, nestin, were noticeably decreased after being treated with GDF11, as compared to control levels (Fig. 2A; p < 0.01). By contrast, the GDF11-treated groups showed significant increase in βIII-tubulin (neuronal biomarker) and GFAP (astrocytic biomarker) mRNA expression as compared to the control (Fig. 2A; p < 0.05). These all indicated the maturation and differentiation of C17.2 neural stem cells. The differences in nestin mRNA expression among the groups of GDF11 treatment were, however, not significant, similar to βIII-tubulin and GFAP. Concomitantly with the mRNA expression, the protein levels of nestin, βIII-tubulin and GFAP confirmed the similar results by western blot (Figs. 2B and 2C). When compared with the control, GDF11-treated cells showed the protein level of nestin was significantly attenuated whereas βIII-tubulin and GFAP were up-regulated (Figs. 2B and 2C), further indicating that GDF11 induced neuronal and astrocytic differentiation. However, no dose-dependent effect of GFD11 was observed.

The results of Annexin V-FITC/PI dual staining revealed that GDF11 substantially induced apoptosis of C17.2 cells. As shown in Figs. 1B and 1E, the number of total (both early and late) apoptotic cells significantly increased in a GDF11 dose-dependent manner. After 72 h of cultivation, the apoptotic cells were negligible in C17.2 cells without GDF11-treated, whereas there were 2.1%, 9.8%, 13.1% and 17.7% of cells exhibiting apoptosis as a result of exposure to 12.5, 25, 50 and 100ng/mL GDF11, respectively (p < 0.05). Meanwhile, the amount of necrotic cells showed a slight but significant increase when treated with GDF11.

GDF11 suppressed the migration of C17.2 cells

The migration of C17.2 cells was performed by a “scratch wound healing” assay. The wound closure data are shown in Fig. 3. It was observed that the wound closure increased as cell migration progressed over time. After 12 h, the wound area had little difference compared to the initial scratch area. As compared with that of 0 h, wound area of 36 h significantly decreased, displaying 25.1% (0 ng/mL GDF11), 64.9% (12.5 ng/mL GDF11), 60.4% (25 ng/mL GDF11), 70.9% (50 ng/mL GDF11) and 75.7% (100 ng/mL GDF11) wound area, respectively (Fig. 3B). These implied wound closure was significantly inhibited when cells were treated with GDF11. Of note, it was revealed that GDF11 showed slight but significant dose-dependent effects in the inhibition of the migration. Together, these results demonstrated that GDF11 significantly suppressed (but not completely abolished) the migratory potential of C17.2 neural stem cells.

Figure 3 GDF11 inhibited the migration of C17.2 cells.

Scratch-wound closure was monitored over time. (A) Representative images showed that GDF11 induced significantly decreased migration speed compared with control (GDF11 untreated cells). Black lines in each graph were pointed toward wound edges. (B) Quantification of the remaining wound area uncovered by migrating C17.2 cells revealed a significant inhibition of migration in GDF11-treated cells. The scratch wound areas at time point 0 hour were set to 100%, and the wound areas at other time point were normalized to their respective 0 hours. Bar is 500 µm (n = 5; *p < 0.05).

GDF11 activated phosphorylation levels of selected signaling kinases

We deduced that, in C17.2 cells, GDF11 transmitted signals through phosphorylation of Smads, as GDF11 belongs to TGF-β superfamily. First of all, we analyzed the effects of GDF11 on TGF-β signal pathway (the classical pathway activated by TGF-β family members) in C17.2 neural stem cells. GDF11 showed no effect on both the mRNA and protein levels of Smad3 (Figs. 2E and 4A). For Smad2, GDF11 significantly up-regulated the transcriptional level other than the protein level (Figs. 2E and 4A). As shown in Fig. 4A, cells untreated with GDF11 (control) displayed negligible phosphorylation of Smad2/3. On the contrary, presence of GDF11 pronouncedly phosphorylated Smad2/3 (p < 0.05). However, no dose-dependent effect of GDF11 on Smad2/3 phosphorylation was observed. Moreover, we investigated the mRNA levels of receptors of TGF-β superfamily, activin type IIB receptor (ActRIIB) and the type I receptors, activin receptor-like kinase 5 (ALK5). The results of qRT-PCR revealed that GDF11 didn’t alter mRNA expression of ActRIIB and ALK (Fig. 2E).

Figure 4 GDF11 increased the phosphorylations of Smad2/3, Creb, p38 and Erk in C17.2 neural stem cells.

(A–D). After 24 h cultivation, GDF11 showed no effect on total protein of Smad2/3, Creb, p38 and Erk, but significantly phosphorylated Smad2 (Ser465/467), Smad3(Ser423/425), Creb(Ser133), p38(Thr180/Tyr182) and Erk (Thr202/Tyr204) in C17.2 cells. N = 4, *p < 0.05, **p < 0.01 and ***p < 0.001 when compared with control.

In order to further research the signal pathways affected by GDF11, we compared the phosphorylation levels of MAPKs in C17.2 cells treated with vehicle or GDF11 using a phospho-MAPK array kit. The fold changes were calculated from the ratio of intensity of the MAPK array from the GDF11 treated C17.2 cells to the control (untreated cells). Cut-off values were set 1.5-fold for up-regulated expression and 0.67-fold for down-regulated expression of a protein. We observed significant increases in the phosphorylation levels of several proteins in GDF11-treated cells compared with the untreated cells (Fig. 5, Table S2 and Fig. S2). Overall, 50% (13/26) of the proteins showed significantly increased phosphorylation after treatment with GDF11, whereas the phosphorylation levels of the remaining 50% (13/26) of the proteins were still unchanged. Strikingly, when treated with GDF11, there were no proteins that showed decreased phosphorylation. In addition, the differentially expressed proteins that showed the most significant increases included Creb (3.42 times increased), HSP27 (3.05 fold increased), Akt1/2/3 (2.55-, 2.47- and 1.50-fold increased expression, respectively), GSK-3β and GSK3α/β (2.12- and 1.50 -fold increased, respectively), p38 α/β (3.21 and 1.73 times increased), Erk1 (1.57), MKK3/6 (2.03- and 1.52- fold increased, respectively) and p70s6k (1.93 times increased) (Fig. 5C and Fig. S2; p < 0.05). These indicated that GDF11 activated the MAPK/Erk and p38 MAPK pathways but not JNK pathway in C17.2 neural stem cells. Remarkably, many of these differentially expressed proteins are involved in signal transductions of cell survival and apoptosis.

Figure 5 Alterations of MAPK pathway-related proteins in GDF11-treated C17.2 cells.

(A) Phosphoproteome profiling of C17.2 cells in response to GDF11 stimulation. Total cell lysates from C17.2 cells with25 ng/mL GDF11- or vehicle-treated were incubated on membranes of the phospho-proteomics platforms (human Phospho-MAPK, 23 different MAPKs and other serine/threonine kinases), as described in “Materials and Methods”. (B) Human Phospho-MAPK array coordinates. (C) The graph shows the relative fold change of proteins with significant difference upon GDF11 treatment, setting 1 for control. Protein levels with higher than ±1.5 folds indicated by dotted lines are considered as the differentially expressed proteins.

Functional classification of differentially expressed proteins

As shown in Table 1, the differentially expressed proteins were mainly classified as cytoplasmic (n = 8), nuclear (n = 7) and mitochondrial (n = 1) proteins.

Table 1 The subcellular location of the differentially expressed proteins.

Proteins	Subcellular location	Fold changes	
GSK-3β	cytoplasm	2.12	
GSK-3α	nucleus	1.50	
CREB	nucleus	3.42	
Akt2	cytoplasm	2.55	
Akt1	cytoplasm	2.47	
ERK1	cytoplasm	1.57	
MKK3	nucleus	2.03	
HSP27	nucleus	3.05	
P38α	cytoplasm	3.21	
p38β	cytoplasm, nucleus	1.73	
p70s6k	nucleus	1.93	
Smad2	mitochondria	1.99	
Smad3	cytoplasm, nucleus	2.01	
Akt3	cytoplasm	1.50	
MKK6	cytoplasm	1.52	

For an overview of the differentially expressed proteins, GO annotation was carried out to identify the significantly enriched GO functions. According to the analysis, the 15 differentially expressed proteins between GDF11-treated cells and control were mainly clustered into 38 functional groups, including 18 biological processes, 12 cellular components, and eight molecular functions (Fig. 6A).

Figure 6 Functional classification and protein–protein interaction of the differentially expressed proteins in GDF11-treated C17.2 cells and control.

(A) According to GO annotation, the differentially expressed proteins between GDF11-treated cells and control were mainly clustered into 38 functional groups, including 18 biological processes, 12 cellular components, and eight molecular functions. (B) Protein domain categories of the differentially expressed proteins were annotated by InterProScan. (C) The protein–protein interaction network of the differentially expressed proteins was analyzed by KEGG) database.

The biological process category according to GO annotations indicated that all of the 15 differentially expressed proteins were involved in the metabolic process. Other significant function groups included cellular process (13/15), response to stimulus (13/15), signaling (13/15) and localization (12/15), etc (Fig. 6A).

In the category of cellular components, the differentially expressed proteins were mainly involved in the cell (13/15), organelle (14/15), cytoplasm (12/15) and nucleus (13/15), indicating the similar subcellular localization that was obtained from WoLF PSORT (Table 1). Not only the similarities but also differences were found between the cellular component category and subcellular localization results. According to the functional analysis of GO annotation, we found six proteins were involved in plasma membrane, however, no membrane-associated proteins were observed from subcellular localization results.

The most representative molecular function category was “binding”, which accounted for all the 15 differentially expressed proteins, and most of the differentially expressed proteins were also involved in catalytic activity (11/15), transferase activity (11/15) and kinase activity (11/15). These results also elucidated that the related signal pathways activated by GDF11.

KEGG enriched pathways

To explore the potential mechanisms for GDF11-mediated cell behavior (cellular proliferation, differentiation, apoptosis and migration) in C17.2 neural stem cells, we used the KEGG database to determine the protein-protein interaction networks and pathways involved in the up-regulated phosphoproteins. The 15 differentially expressed proteins were mainly mapped to 51 pathways according to the KEGG database, which were mainly associated with environmental information processing (signal transduction), organismal systems (immune system, nervous system, endocrine system and ageing), cellular processes (cell growth and death, transport and catabolism, and cellular community) and human diseases (drug resistance, endocrine and metabolic diseases, neurodegenerative diseases, infectious diseases, and cancers) (Table S3). These all indicated the differentially expressed proteins were mainly involved in signal transduction of cellular behavior. Furthermore, domain functional description of the differentially expressed proteins annotated by InterProScan, were significantly enriched in protein kinase domain (25.58%) and protein kinase-like domain (25.58%) (Fig. 6B). These were also in line with the results of molecular function category of GO annotation which indicated the differentially expressed proteins were mainly in connection with catalytic activity, transferase activity and kinase activity (Fig. 6A).

A major overlapping network was enriched in this analysis (Fig. 6C). Three canonical signaling pathways (TGF-β, PI3K-Akt and MAPK signaling pathways), that were activated by the up-regulation of phosphoproteins, were identified, and the cross-talking signaling cascade was shown as well. One mainly functional cluster was apparent in the protein-protein interaction, including Akt1/2/3 (the key components in the PI3K-Akt signal pathway), Erk1 (the key components of MAPK/Erk pathway) and p38α/β (the key components in the p38 MAPK signal pathway). These results provided a possible resource for future studies of the proteins involved in GDF11-treated C17.2 cells.

Validation of selected differentially expressed proteins

To confirm the results of phospho-MAPK array, three differentially expressed protein candidates (Creb, p38 and Erk) were selected for further validation using western blot. Total protein lysates from C17.2 cells cultured with indicated concentrations of GDF11 (0, 12.5, 25, 50 and 100 ng/ml, respectively) were prepared and the phosphorylation levels were determined by their respective phosphorylated antibodies. When compared with control, no detectable changes in total Creb, p38 or Erk protein expression were observed in GDF11-treated C17.2 cells. Nevertheless, GDF11-treatment significantly increased the phosphorylation levels of Creb, p38 and Erk (Figs. 4B, 4C and 4D; all p < 0.05). These western blot results were generally consistent with the results of the phospho-MAPK array.

Discussion

Around the world, the number of aged people is precipitously increasing. Therefore, searching for anti-ageing or rejuvenating factors is quite important to develop therapeutic strategies for the treatment of age-related diseases. Recently, GDF11 was suggested as a potential rejuvenating factor, which not only reversed age-related cardiac hypertrophy and dysfunction in skeletal muscle in mouse (Loffredo et al., 2013; Poggioli et al., 2016; Sinha et al., 2014), but also induced rejuvenation of impairments in cognitive function of ageing mouse by remodeling the cerebral vascular and enhancing neurogenesis (Katsimpardi et al., 2014). Recently, a prospective cohort study also revealed that higher levels of GDF11/8 were associated with lower risk of cardiovascular events and death in patients with stable ischaemic heart disease (Olson et al., 2015). However, these initial findings have been challenged by later recent studies. It was reported that GDF11 inhibited muscle regeneration and decreased satellite cell expansion in mice (Egerman Marc et al., 2015). Hinken et al. (2016) also suggested GDF11 wasn’t a rejuvenator for aged murine skeletal muscle satellite cells. In addition, restoring GDF11 in old mice showed no effect on pathological hypertrophy (Smith et al., 2015).

Because GDF11 was reported to improve neurogenic rejuvenation, we hypothesized that GDF11 influenced the cellular behavior of C17.2 neural stem cells, including viability, proliferation, differentiation, apoptosis and migration. Therefore, we focused on the effects and potential mechanism of action of GDF11 on viability, proliferation, differentiation, apoptosis and migration in C17.2 cells. Here, our results indicated that GDF11 substantially induced differentiation and apoptosis, and suppressed migration of C17.2 cells mainly through MAPK signal pathway. Meanwhile, GDF11 induced sight but significant increases in cellular viability in a short time of growth (24 h) and showed no effects on cellular viability for medium-term cultivation (<4 passages; approximately 10 days). For long-term cultivation (>4 passages), high concentrations of GDF11 significantly inhibited the proliferation of C17.2 cells (Figs. 1C and 1D; p < 0.05). To the best of our knowledge, we are unaware of any similar published results. Strikingly, similar to our results, it was found that GDF11 slightly increased cell viability after short-term treatment and slightly decreased cell viability after long-term treatment in human umbilical vein endothelial cells (Zhang et al., 2016). Previously, it was suggested that GDF11 acted as a negative regulator of neurogenesis (Wu et al., 2003). Recently, Williams et al. (2013) reported a controversial finding that GDF11 suppressed proliferation and migration of Cor-1 cells, whereas no effect on differentiation was observed. The conflicting results may be caused by GDF11 from different vendors, different batches of GDF11 from the same manufacturer, or cells from different sources.

As a member of TGF-β superfamily, it was reported that GDF11 activated TGF-β signal pathway as a consequence of phosphorylating Smad2/3 in several cell types (Liu et al., 2016; Loffredo et al., 2013; Zhang et al., 2016). In the present study, we successfully observed GDF11 phosphorylated Smad2/3 in C17.2 neural stem cells. Consistent with our results, it was also confirmed that Cor-1 neural stem cell line was able to respond to GDF11 stimulation by Smad2/3 phosphorylation (Williams et al., 2013). It is widely known that the Smad2/3-dependent TGF-β signals have been implicated in the proliferation and differentiation of neural stem cell. For example, GDF11 negatively regulated self-renewal of neuroepithelial stem cells through TGF-β signals (Falk et al., 2008).

Proteins, not genes, are the specific practitioners of cellular life organisms. Although genome and transcriptome analyses are very useful to reveal the mechanism of GDF11 stimulation, proteomic profiles may not be accurately predicted by transcriptome profiling due to several factors, such as post-translational modifications. Therefore, research on proteomics is helpful to provide new information concerning the C17.2 cells response to GDF11 stimulation. Based on phospho-proteome profiling array and bioinformatic analysis (Figs. 5 and 6), we found 15 differentially expressed proteins, including p38, Erk, Akt, GSK3α/β, Creb, MKK3/6, p70S6k and HSP27, which were mainly involved in signal transductions of cell survival and apoptosis. Besides TGF-β signal pathway, we also found Akt pathway and two important MAPK pathways (Erk MAPK and p38 MAPK pathways) were activated, but not JNK pathway. Similarly, it also reported that GDF11 activated TGF-β/Smad2/3 but suppressed JNK signaling pathways in apolipoprotein E-null mice (Mei et al., 2016). The functions of Erk MAPK and p38 MAPK pathways are complex, which are involved in controlling cell proliferation, differentiation, survival/apoptosis and migration (Wagner & Nebreda, 2009). Various studies demonstrated Erk MAPK pathway was involved in cellular proliferation and migration (Khodosevich, 2009; Wu et al., 2014). Although p38 MAPK pathway is normally associated with anti-proliferative and apoptotic functions (Wagner & Nebreda, 2009), it was also reported that p38 was implicated in pro-survival functions, including positively regulating proliferation, differentiation and anti-apoptosis (Halawani et al., 2004; Ricote et al., 2006; Terriente-Félix et al., 2017; Thornton et al., 2008). MAPK/Erk, Akt and p38 MAPK pathways were required for the migration of cortical neurons upon HGF stimulation (Segarra et al., 2006), however, we observed GDF11 significantly suppressed the migratory capacity of C17.2 neural stem cells with the activation of Erk MAPK, PI3K/Akt and p38 MAPK pathways. Of the 15 differentially expressed proteins we identified, HSP27 and p70S6K are two important downstream effectors of Akt pathway. Mechanistically, PI3K/Akt phosphorylates HSP27 and p70S6K, which facilitate protein folding and control protein synthesis, to inhibit apoptosis and promote proliferation (Khodosevich, 2009; Li et al., 2008; Rane et al., 2003). All of these suggested that GDF11 regulated the proliferation, differentiation, apoptosis and migration of C17.2 cells by cross-talking with MAPK signaling pathway.

Neuronal migration is a complex and key process in physiological and pathological conditions. Increasing the quantity of nerve cells and the migration of neurons to the final position are critical to reverse age-related dysfunction in brain (Contreras-Vallejos, Utreras & Gonzalez-Billault, 2012; Martino et al., 2011; Zhao, Deng & Gage, 2008). It should be noted that, although GDF11-treatment for 24 h slightly increased the viability of C17.2 cells (Fig. 1C), it showed no effect on cell death (Fig. S1B ). GDF11 didn’t change the cell viability after 72 h cultivation (Fig. 1C), whereas it significantly stimulated cell death (Fig. S1A). In addition, we found GDF11 significantly suppressed the migration of C17.2 cells. Despite the fact that GDF11 indeed induced C17.2 cells to differentiate into neurons and astrocytes, our point of view is that it should be cautious if GDF11 is considered as a rejuvenated factor for neural stem cells.

Conclusion

In the present study, we found that GDF11 was an important regulator of neural stem cell. In C17.2 neural stem cells, GDF11 showed a positive effect on cell viability after 24 h treatment but displayed a tendency of a negative effect for long-term cultivation. In addition, GDF11 significantly induced differentiation and apoptosis, and suppressed migration of C17.2 neural stem cells. Further analysis of MAPK signaling pathway, which was activated by GDF11, preliminary illustrated the potential mechanism of action by which the cellular behavior was induced. Taken together, our current findings implied that GDF11 might be a potential target for pharmacologic blockade instead of a rejuvenated factor for neural stem cells.

Supplemental Information

Table S1 Primer sequence for qRT-PCR

Click here for additional data file.

Table S2 The details of the Human Phospho-MAPK Array coordinates

Click here for additional data file.

Table S3 KEGG pathway enrichment of differentially expressed proteins

Click here for additional data file.

Figure S1 The morphology of C17.2 cells treated with indicated concentrations of GDF11

Images were obtainedat 50Xmagnification by inverted fluorescence microscope. The live cells were stained with calcein AM in green, and the dead cells were stained with EthD-1 in red.

Click here for additional data file.

Figure S2 The overview of the 23 MAPKs and other serine/threonine kinases detected using human Phospho-MAPK array kit

Click here for additional data file.

Data S1 Raw data for statistical analysis

Click here for additional data file.

Data S2 Raw data for Western blots

Click here for additional data file.

The authors are grateful for helpful comments of bioinformatic analyses from Jingjie PTM BioLab (Hangzhou) Co. Ltd (China).

Additional Information and Declarations

Competing Interests

Author Contributions

Data Availability

The authors declare there are no competing interests.

Zongkui Wang conceived and designed the experiments, performed the experiments, analyzed the data, prepared figures and/or tables, authored or reviewed drafts of the paper, approved the final draft.

Miaomiao Dou performed the experiments, authored or reviewed drafts of the paper, approved the final draft.

Fengjuan Liu performed the experiments, approved the final draft.

Peng Jiang performed the experiments, analyzed the data, contributed reagents/materials/analysis tools, prepared figures and/or tables, approved the final draft.

Shengliang Ye performed the experiments, analyzed the data, prepared figures and/or tables, approved the final draft.

Li Ma analyzed the data, contributed reagents/materials/analysis tools, approved the final draft.

Haijun Cao, Pan Sun, Na Su and Fangzhao Lin contributed reagents/materials/analysis tools, approved the final draft.

Xi Du performed the experiments, approved the final draft.

Rong Zhang and Changqing Li conceived and designed the experiments, authored or reviewed drafts of the paper, approved the final draft.

The following information was supplied regarding data availability:

The raw data are provided in the Supplemental Files.

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
