# Peer review of "GDF11 induces differentiation and apoptosis and inhibits migration of C17.2 neural stem cells via modulating MAPK signaling pathway"

_PeerJ, doi:10.7717/peerj.5524_

## Round 0.1 · original submission · Minor Revisions

Please address the authors comments and respond to them.

1- Manuscript needs extensive proofreading. You should pay attention to spaces between words and brackets.
2- In-text citations require attention : for example you can't use the format: Egerman and colleagues(Egerman et al. 2015). You either use Egerman et al. (2015) or cite the source at the end.
Another example : Hinken et al.(Hinken et al. 2016). Correct this format: Hinken et al. (2016).
3- Page 34 remove and so on
4- These data demonstrated the effects of GDF11 on neural stem cell through the MAPK pathway : What are these effects?
5- Rewrite the following sentence: GDF11 induced cellular proliferation in a short time, however it inhibited proliferation in a long-term cultivation.
6- Phosphorylation does have s
7- You should provide the catalogue numbers for all materials used especially antibodies. Source of the MAPK array and catalogue number.
8- Should use Standard error of the mean not standard deviation
9- Page 12 remove etc at the end of the page
10- Provide the missing figure (see reviewers comments).

Reviewer 1 ·

Basic reporting

The overall quality of the text needs improvement, especially concerning English writing and syntax. E.g line172, 411, 414, 418…
The structure of the manuscript is fine and follow the journal requirements.

The raw data for the ERK WB (Figure 4d) is missing.

Experimental design

The experiments fit and follow a logical thought based on the initial hypothesis. It fits within the debate on the potential role of GDF11.
The experimental section is clear enough to allow researchers to reproduce the experiments.

Validity of the findings

The results support a new possible application for GDF11 as therapeutic target which is a new input in the field. It brings some new arguments in the current debate on the role of GDF11 as anti-aging agent.
The data are robust and conclusion drawn according to the findings.

Additional comments

The paper shows the involvement of the GDF11 in neural cell differentiation and apoptosis, the experiments and results support the initial hypothesis raised by the authors. The link with the MAPK pathways have been well highlighted and more generally, this work opens new questions about contradictory role of this protein on anti-aging.

Reviewer 2 ·

Basic reporting

 The overall quality of the text is not poor, but there is space for improvement, especially concerning the correct use of the English language. In addition, some vague, not clear wording has been used in some instances (e.g. line 86, 144,152,156-159, 198,230-232,270,316,411,427,489 etc.)- further proof-reading is suggested.
 The overall structure of the manuscript is generally in accordance with the basic suggested structure, however, some results are presented in the Introduction section. In addition, the background given could be richer.
 Section 2.3 is a bit too dense- it could be broken down into smaller sub-sections.
 The supplementary tables are complete and clear.
 The supplementary figures are generally up to a high-quality standard; however, the Erk1/2 for Fig.4d supplement is missing

Experimental design

 Overall, the experimental processes used throughout the study are clearly presented in the ‘Materials and Methods’ section and are analytical enough for other researchers to reproduce the work.
 In addition, there is a logical sequence of the experiments, which was followed in order to address the hypothesis and shed light on the controversy over GDF11.
 However, I believe two points should be carefully addressed:
o The confluence of the cells for the scratch assay may need to be higher, even ~100%
o In the statistical analyses, it may be a better approach to present cumulative results as mean +/- SEM

Validity of the findings

 The results of the study are well-presented, organised and seem to be robust, given the statistical analysis presented. This is an in-depth study of the effects of GDF11, which addresses one by one numerous aspects of functional effects and answers a number of different components presented in the hypothesis.
 However, for aesthetic purposes, in the scratch assay results, the use of straight lines to approximately indicate the borders seems a better solution than the curved lines.
 Conclusions have been rationally drawn based on the study’s findings.

Additional comments

Overall, this is a robust and integral piece of work. The group have rationally designed and executed the current study, which has successfully provided some important insight on the role of GDF11 on various aspects of the neural cell life and has, as well, added to the controversy over this specific factor. Therefore, there is indeed some contribution of the study to the field. However, the manuscript itself needs some improvements, especially concerning vocabulary and word structure.

---

## Round 0.2 · Minor Revisions

Please address the comments in the attached document.

---

## Round 0.3 · accepted · Accept

I am pleased to inform you that your paper is accepted for publication. Thank you for addressing the comments and correction.

#